# Identification of potential vulnerable points and paths of contamination in the Dutch broiler meat trade network

Shuai Hao[1], Ayalew Kassahun[2]*, Yamine Bouzembrak[3], Hans Marvin[3]

**1** Food Quality and Design, Wageningen University, Wageningen, The Netherlands, **2** Information Technology, Wageningen University, Wageningen, The Netherlands, **3** Wageningen Food Safety Research, Wageningen, The Netherlands

* ayalew.kassahun@wur.nl

**Data Availability Statement:** All data used to reach the conclusions are publicly available. The methods section contains detailed instruction where the data are obtained. The data is submitted as a

## Abstract

The poultry meat supply chain is complex and therefore vulnerable to many potential contaminations that may occur. To ensure a safe product for the consumer, an efficient traceability system is required that enables a quick and efficient identification of the potential sources of contamination and proper implementation of mitigation actions. In this study, we explored the use of graph theory to construct a food supply chain network for the broiler meat supply chain in the Netherlands and tested it as a traceability system. To build the graph, we first identified the main actors in the supply chain such as broiler breeder farms, broiler farms, slaughterhouses, processors, and retailers. The capacity data of each supply chain actor, represented by its production or trade volumes, were gathered from various sources. The trade relationships between the supply chain actors were collected and the missing relationships were estimated using the gravity model. Once the network was modeled, we computed degree centrality and betweenness centrality to identify critical nodes in the network. In addition, we computed trade density to get insight into the complexity of subnetworks. We identified the critical nodes at each stage of the Dutch broiler meat supply chain and verified our results with a domain expert of the Dutch poultry industry and literature. The results showed that processors with own slaughtering facility were the most critical points in the broiler meat supply chain.

## Introduction

In the past decades, the poultry meat market share has increased globally due to its high nutritive value, low cost and low-fat content [1, 2]. The Netherlands is one of the largest global producer of poultry meat [3] with a total poultry meat production reaching over 1.1 million tons in 2018 [4]. The vast majority of poultry meat produced is broiler meat [5]. Although the broiler meat products are preferred by an increasing number of consumers, it has been implicated as a source of food-borne diseases for humans; particularly, due to the presence of pathogens, like *Salmonella*, *Campylobacter* and *Listeria monocytogenes* [6, 7]. In 2017, the highest

Supporting Information files in a zip file called data. zip. In addition the available data is publicly available at: https://github.com/akassahun/dutch_broiler_meat_network A small record of the interview text described in the methods section is not included. However, the trade data information that was obtained through interviews are included. We are not authorised to share the interview text, but on the other hand, the interview text is IN NO WAY part of the minimal dataset, and was not needed to reach the conclusions.

**Funding:** The author(s) received no specific funding for this work.

**Competing interests:** The authors have declared that no competing interests exist.

recall of broiler meat in the United States was because of the presence of pathogens in the meat [8].

The spread of a contamination is difficult to control because products are moving quickly among the various actors in a supply chain, as observed for the broiler meat supply chain [9, 10]. Evidently, efficient traceability systems covering whole supply chain are essential for quickly identifying the origin and the extent of the contamination [11]

A traceability system can be based on the trade network structure as demonstrated by Natale *et al.* [12] and Lebl *et al.* [13]. Generally, the actors with large number of trade contacts in a trade network are more vulnerable in spreading diseases and contaminations. These actors are sometimes referred to as *'central'*. If an actor has many trade relations with several suppliers, it is prone to receiving contaminated food and vice versa. The potential sources and paths of contamination can, therefore, be identified quickly by analysing the trade network [14]. A trade network can formally be represented and visualized as a graph, which contains a set of vertices (i.e. actors in the supply chain) and edges (i.e. trade relationships). Ideally, the graph is modelled by identifying each individual actor in the trade network, determining the actor's suppliers and customers, and representing the volumes of the trade between all actors in the supply chain. However, in practice, such detailed information is often missing. Not all actors are known individually, and those that are known do not generally reveal their trade relations and trade volumes. To overcome this problem gravity models can be used to estimate the links among the nodes using some proxy data such as capacities of the actors and the geographical distances among them [14–16]. More details about the Graph theory is provided in the supplement.

The aim of this paper is to explore the suitability of graph theory as a traceability system and used the broiler meat supply chain in the Netherlands as a proof of principle. We demonstrate how to determine the nodes of the network and to quantify the centrality of the nodes with graph theory using two indicators for centrality: degree centrality and betweenness centrality. In order to obtain insight into the complexity of the entire network, sub-networks were modelled and the trade densities were computed as a third indictor. Critical nodes at each stage of the Dutch broiler meat supply chain were identified which were confirmed by a domain expert.

## Materials and methods

### Data collection

The following information is needed to develop a graph network: 1) the identity of the actors or group of actors that can be modelled as nodes, 2) their production capacity or trade turnover, and 3) their geographical locations. The Dutch broiler meat supply chain network consists of the following actors: the breeding organizations (parent stock breeder), breeder farms, hatchery firms, broiler farms, slaughterhouses, processors, retailers, consumers, feed producers, importers of live broilers, exporters of live broilers, importers of meat and exporters of meat.

In modelling the graphs, breeding organizations were omitted in the modelling because parent stock breeder firms are unlikely sources of contaminations. No distinction is made between hatchery and breeding farm nodes, since most breeders in the Netherlands have hatchery facilities—and vice versa. Finally, since no data is available about retailers and individual consumers, the consumer populations of municipalities were used as consumers nodes. The supply chain we modelled thus consists of seven different types of actors, which are: *breeder farms*, *broiler farms*, *slaughterhouses*, *processors*, *retailers/consumers*, *importers*, and

**Table 1. Data sources.**

| Data source | Actor type | Link |
|---|---|---|
| Banach et al. | Breeders | Chemical and Physical Hazards in the Dutch Poultry Meat Chain [17] |
| CBS Statline | Broiler breeder and breeder farms | Agriculture; crops, livestock and land use by general farm type, region [18] |
| Van Plaggenhoef | Broilers | Integration and self regulation of quality management in Dutch agri-food supply chains [19] |
| Kompass | Slaughterhouses and processors | Global B2B Online Directory [20] |
| D&B Hoovers | Slaughterhouses and processors | Company search [21] |
| Statista | processors | Number of enterprises in the processing and preserving of poultry meat industry in the Netherlands from 2008 to 2016 [22] |
| The Poultry Site | Slaughterhouses | Poultry Meat in the Netherlands [23] |
| Interview | Slaughterhouses and processors | n/a |
| PPE | Hatchery | PPE annual report on poultry meat and eggs [24] |
| Retail Insiders | Wholesalers | Poulterers [25] |
| CBS | Inhabitants/consumers | Population data from the Dutch Central Bureau or Statistics [26] |
| Statista | Consumers | Per capita consumption of poultry in the Netherlands from 2007 to 2018 [27] |

*exporters*. From these actors, trade data was collected, mainly capacities and locations, using the data sources listed in Table 1.

Ideally, individual breeding and broiler farmers should be present as nodes in our graph model but unfortunately no information on this level was available for this study. To overcome this problem, we used instead aggregated data at the municipality level in the same manner as consumers data. This means, all breeder farms in one municipality were aggregated into a single node; likewise, all broiler farms, final consumers were aggregated at municipality level. We refer to breeder farms, broiler farm and consumer nodes as 'municipality nodes'. On the contrary, the required data about the individual slaughterhouses and processing facilities was available; therefore, these actors were modelled as individual nodes. Trade data of individual importers and exporters were lacking; therefore, all importers were aggregated into one single node and all exporters were aggregated into another single node.

Aggregated trade data of broiler breeder farms, broiler farms and consumers, including the number of farms in each municipality, was obtained for the year 2018 from CBS Statline. In addition, the capacities (number of chickens) of breeder farms in each municipality for the same year was retrieved for this data source. We obtained trade and location data of each individual slaughterhouses and processors from Kompass and D&B Hoovers for the calendar year of 2018. The data showed that some of the businesses have both slaughtering and processing facilities which we hereafter refer to as either "processor with a slaughterhouse" or "slaughterhouse with a processing facility". The processors with slaughtering business buy live chicks from broiler farms and sold packaged products to retailers. We obtained the capacities of some slaughterhouses (kilograms of broiler meat they slaughtered per week) for the calendar year of 2010 from The Poultry Site (Table 1). Finally, missing and additional trade information was collected through telephone interviews with 16 slaughterhouses and 12 processors. For the processing plants, their capacities are assumed to be proportional to the turnover. Hence, the capacities of processors were replaced by their turnover obtained from D&B Hoovers and Kompass. The number of inhabitants of each municipality for 2018 was obtained from CBS (Table 1).

## Modelling the broiler meat trade network

After data collection, the visualization network was created using QGIS software [28]. To identify the critical nodes scientifically and precisely, a quantitative data analysis was performed to determine the degree centrality and betweenness centrality of each node. Then trade density was calculated to analyse the edge density networks. The degree centrality was simply calculated by using sum of total in-coming and out-going arcs of the nodes in the network. Also, the histograms figures were created to analyse the distribution of in-degree, out-degree centrality. The input data was the trade relation between two nodes in the network. The trade density was also calculated with total number of nodes and edges in each subnetwork. After data analysis, the centrality of each actor was measured to determine where the critical actors are.

## Results and discussion

### The structure of the Dutch broiler meat supply chain

In the Netherlands, the main actors in broiler meat supply chain are: breeding organizations, breeder farms, hatchery firms, broiler farms, slaughterhouses, processors, retailers, consumers, feed producers, importers of live broilers, exporters of live broilers, importers of meat and exporters of meat. Their interrelationships are summarized in Fig 1 and described below.

The Dutch broiler meat supply chain starts with *breeding organizations*, which stock grandparents of broilers. The two main breeding companies providing most of the broiler parent chickens (also called breeders) are: Aviagen and Cobb-Vantress Inc. The third breeding company is Hubbard, which provides slow-growing chickens [17]. Broiler parents, which produce hatching eggs, are stocked in broiler breeder farms. In 2013, there were 640 breeder farms in the Netherlands with the capacity of 6 million breeders. The breeders produced 1 billion hatching eggs, which hatch to become broilers. The eggs are hatched in *hatching firms*. In 2013, there were 17 hatching firms which hatched 650 million eggs per year [24]. Most of the breeder farms have their own hatching centres. Thus, breeder farms also deliver chicks to broiler farms

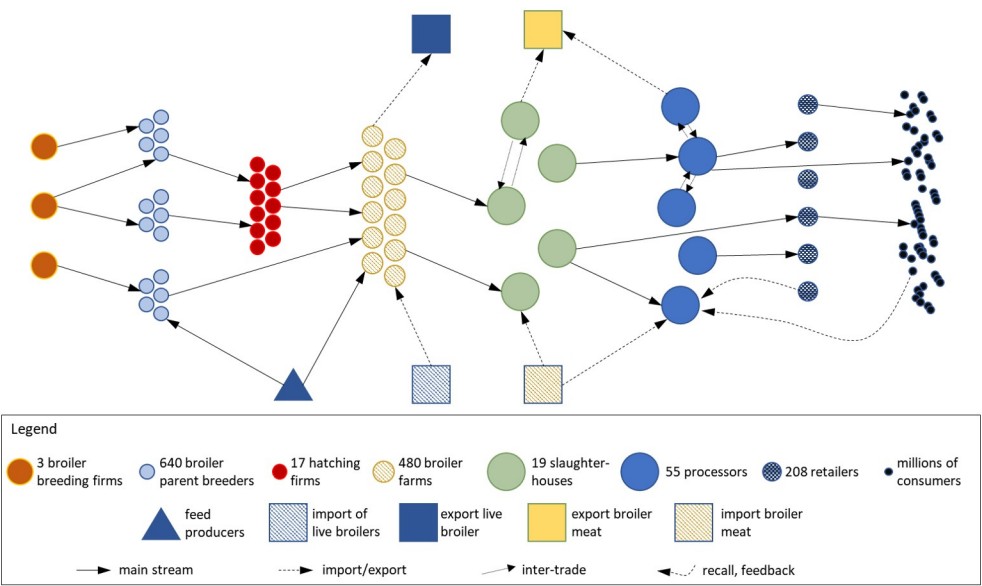

**Fig 1. Schematic representation of Dutch broiler meat supply chain network.**

directly. Half of the broilers chicks were intended for Dutch broiler farms and the other half were exported to other countries [17].

After the chicks are hatched, the day-old chicks are transported to *broiler farms*, where the chicks are kept for different time periods depending on the target market. Five to seven weeks-old broilers are sold as conventional produce; eight weeks-old broilers are sold as animal-friendly produce; and organic broilers are sold at the age of ten weeks [17]. The number of the Dutch broiler farms has decreased significantly, from 2329 in 2007 to 480 in 2013 [19, 24]. However, the capacity (total number of broilers) of each farm increased because farms have risen their production capacity significantly. In 2000, the average production of broilers per broiler farm was 46,560, which increased to 77,180 by 2018 [19, 24, 29]. Broilers are sold to slaughterhouses with the live weights between 1.8kg to 2.6kg. There are only a limited number of broiler slaughterhouses; according to the 2013 data there are just 19 slaughterhouses. They processed 1.14 million tons of live weight broilers and produced 0.84 million tons of slaughtered weight [24]. Some of the broilers are slaughtered in Germany and Belgium and the semi-finished products are transported to Dutch processors. After cutting, the broiler meat is delivered to *processors* for packaging and labelling [17]. In 2016, there were 55 processors in the Netherlands, a slight increase from 50 in 2008 [22]. There are also inter-trade linkages among slaughterhouses and among processers to balance the over- and under-supplies. For example, slaughterhouses can deliver slaughtered broiler meat to other slaughterhouses to guarantee sufficient supply to broiler processors. Then, packaged products are delivered to *retailers*. In 2017, the number of Dutch broiler stores reached 208, amounting to 2% of the 13,297 specialized stores. Finally, raw broiler meat is sold to *consumers*, including private and commercial bulk consumers, such as restaurants. Private consumers buy raw poultry meat from supermarkets (79%), butcheries (13.6%), open markets and streets (4.0%), other food specialized stores (1.2%) and other channels (2.2%) [25]. The per capita consumption of poultry meat is over 22kg per year and per capita consumption of broiler meat can reach 18kg per year [27]. There are also reverse trade connections from downstream (consumers) to upstream (farms) in the supply chain. For example, some by-products processed during production would be transported back to farm as animal feeds. The recall of unqualified products also belongs to the reverse trade connection [14].

Besides these main actors, several other actors are involved in the broiler meat supply chain. First in this category are poultry feed producers, on whom breeder and broiler farms depend for high quality feed. Second are importers and exporters of live chicks. In 2013, the Netherlands imported 185,000 tons and exported 42,500 tons of live birds from and to other EU countries. The third are importers and exporters of slaughtered chicken and meat. The Netherlands imported 405,000 tons of frozen broiler meat from Thailand, Brazil, Belgium and Germany and exported 943,000 tons of frozen broiler meat to other countries [17]. The main export markets for Dutch broiler meat are Germany and the U.K., where 30.6% and 19.1%, respectively, of broiler meat is exported to [19].

The data collected as described above is used to construct a network graph of the Dutch broiler meat supply chain. Ideally, the individual actors and their trade relations or trade volumes should be known to construct such a network graph. However, such data was not always available and, therefore, as described in the introduction we aggregated some actors by municipality so that we can use trade volume data that is publicly available. The resulting network graph of such grouping consisted of 245 broiler breeder farms situated in 90 municipality nodes; 531 broiler farms situated in 162 municipality nodes; 20 broiler slaughterhouse nodes; 41 broiler processor nodes (including 17 processors with slaughtering business and 24 processors without slaughter business); and almost n = 17 million end consumers (retailers) situated in 380 municipality nodes as shown in Fig 2.

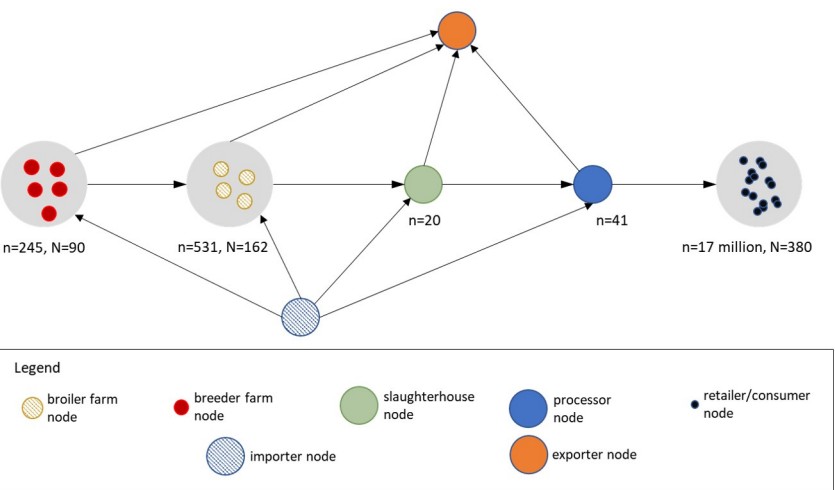

**Fig 2. Schematic representation of the analysed broiler meat supply chain.**

To be able to estimate the trade relationships between the nodes using the gravity model, a scaling factor $q_0$ is used. This factor was determined iteratively. Firstly, a random value of $q_0$ was selected and a draft network is generated. The draft network estimates the trade relation for all nodes. Then, the trade relations generated are matched with the few known trade relations in terms of the number of suppliers and customers. If the trade relations match $q_0$ is adjusted a new draft network is generated. This process continues until the trade relations estimated by the generated network match the known trade relations. For example, we know from the data that one breeder farm located in Boxmeer sold products to 10 municipalities in the breeder farm-broiler farm sub-network. The scaling factor that match this and other known trade relations was found to be $q_0 \approx 1/2426$.

## Broiler meat trade network

Fig 3 shows the Dutch broiler meat trade network estimated by the gravity model. The trade network comprised a total of 693 nodes and 9,285 edges in 2018, including 245 broiler breeder farms situated in 90 municipality nodes, 531 broiler farms situated in 162 municipality nodes; 20 broiler slaughterhouses nodes; 17 broiler processor with a slaughterhouse nodes; 24 broiler processor without a slaughterhouse nodes and retailers (consumer) nodes situated in.

The entire network consists of seven groups of nodes: broiler breeder farms nodes, broiler farms nodes, slaughterhouses nodes, processors nodes, retailers (final consumers) nodes, importer nodes and exporter nodes. The processors nodes (dark blue nodes) were surrounded by retailer nodes (orange nodes) and the out-going arcs of processor nodes are in redial patterns. Therefore, processor nodes can be seen as central actors and they are usually critical in broiler meat supply chain. For a deeper analysis, the entire trade network as shown in Fig 3 was decomposed into five sub-networks and are shown in Fig 4. The sub-networks visualize the trade links between two consecutive actors of the broiler meat supply chain.

The sub-network showing the trade links between breeder farms and broiler farms (Fig 4A) appears to be a decentralized network. Most nodes had only one trade relation with others, because the number and distributions of breeder farms and broiler farms were almost the same. They can trade products only with adjacent regions to save the cost and guarantee the

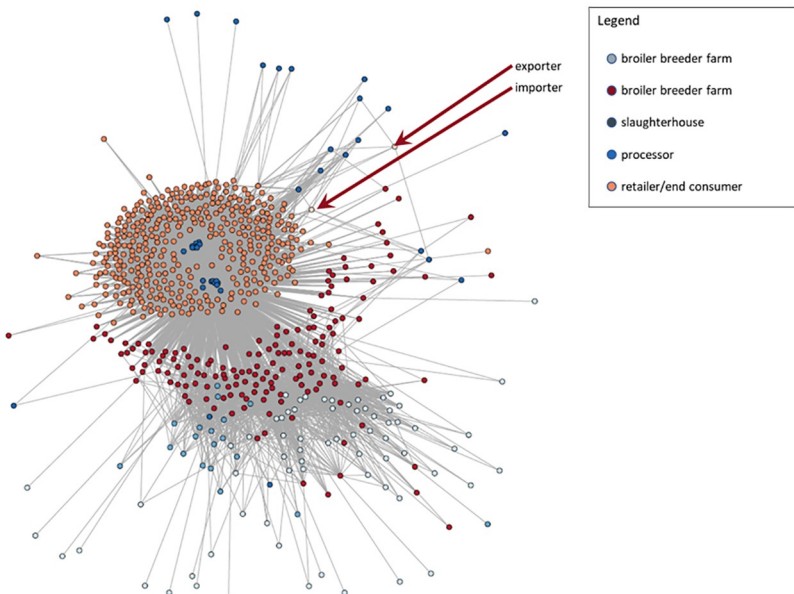

**Fig 3. Visualization of Dutch broiler meat trade network (the position of nodes is not geographic location of actors).**

delivery time. Fig 4B shows the trade links between broiler farms and slaughterhouses; the sub-network appears to be a centralized network and consists of several star structures. The central nodes were slaughterhouse and the satellite nodes were broiler farms. Many broiler farms provided service to a single slaughterhouse. Obviously, slaughterhouses are easily vulnerable to receiving contaminated products. The sub-network (Fig 4C) showing trade links between broiler farms and processors with slaughtering business is also a centralized network with processors as star nodes and broiler farms as satellite nodes. The processors had slaughter facilities and purchased live chicks form a larger number of broiler farm nodes. Like slaughterhouses in the previous sub-network, here the processors are prone to receiving contaminated products. The sub-network between slaughterhouses and processors (Fig 4D) is largely a decentralized network, apparently for the same reason as the sub-network involving breeder and broiler farms. The sub-network showing trade links between processors and retailers/end-consumers (Fig 4E) is a centralized network with processors as star nodes and retailers as retailers as satellite nodes. Each processor sold products to many retailers and end-consumers; therefore, processors are prone to spread contaminated products.

## Vulnerability of actors to spread contaminations

The vulnerability to receive or spread contaminations that have entered the supply chain can be determined by degree centrality, betweenness centrality, and trade density and will be discussed below.

**Degree centrality.** Pinior *et al.* [14] illustrated that the nodes with high in-degree centrality are called "vulnerable". These nodes have higher probability to receive cognations. The nodes with high out-degree are called "virulent", which means if pathogens appear in virulent nodes, they will spread to many consumers. Number of nodes, number of trade relations, in- and out-degree centrality of each actor in the broiler meat supply chain are shown in Table 2.

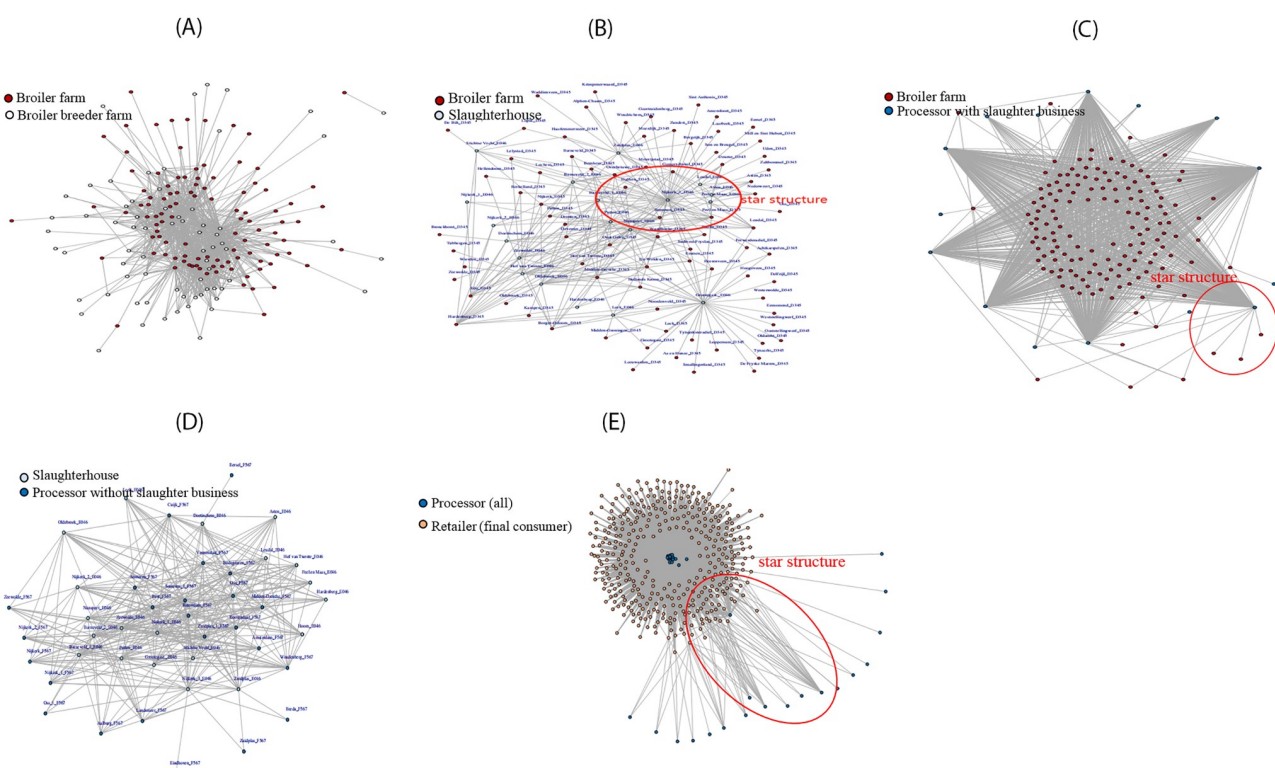

**Fig 4. Visualization of sub trade networks: A) broiler breeder farm-broiler farm; B) broiler farm-slaughterhouse; C) broiler farm-processor with slaughterhouse; D) slaughterhouse-processor without slaughterhouse; E) processor-retailer.**

*Breeder broiler farms and broiler farm*. The in- and out-degree centrality of breeder broiler farms and broiler farms are relatively low. The maximum out-degree centrality of a breeder farm was 85 (of municipality: Someren) and the median out-degree was 3. The maximum in-degree centrality of a broiler farm nodes was 43 (of municipality: Someren) and the median in-degree was equal to 3. The maximum out-degree centrality of a broiler farm node was 25 (*i. e.* Someren), and the median out-degree was 7. The reason for low centrality of the farms is that they most likely receive or deliver chicks from adjacent areas. The computation (the details of which is not included in Table 2) also shows that many of the farms have a single supplier or consumer: 39% broiler breeder farms nodes sold chicks to a single broiler farm node,

**Table 2. In-degree, out-degree centrality and distribution of each actor.**

| Items | | Broiler breeder farm | Broiler farm | Slaughterhouse | Processor with a slaughterhouse | Processor without a slaughterhouse | Importer | Exporter | Retailer/final consumer |
|---|---|---|---|---|---|---|---|---|---|
| **Nr. nodes** | | 90 | 162 | 20 | 17 | 24 | 1 | 1 | 380 |
| **Nr. trade relations** | | 1038 | 2325 | 638 | 3805 | 4133 | 28 | 11 | 6592 |
| **In-degree** | Max | - | 43 | 51 | 146 | 20 | 1 | - | 30 |
| | Median | - | 3 | 11.5 | 16 | 17 | 1 | - | 18 |
| | Min | - | 0 | 0 | 0 | 1 | 0 | - | 1 |
| **Out-degree** | Max | 85 | 25 | 23 | 377 | 378 | - | 1 | - |
| | Median | 3 | 7 | 19 | 53 | 17.5 | - | 1 | - |
| | Min | 0 | 0 | 12 | 0 | 0 | - | 0 | - |

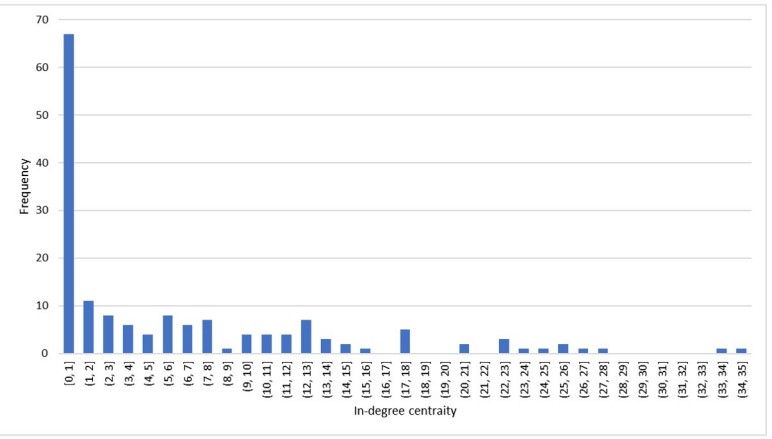

**Fig 5. Distribution of in-degree centrality of broiler farms.**

42% broiler farms nodes received chicks from a single breeder farm, and 72% broiler farms nodes sold live chicks to single slaughterhouse node. The distribution of out-degree centrality of broiler breeder farm, in-degree centrality of broiler farms and out-degree centrality of broiler farms were the same. The distribution of in-degree centrality of broiler farm is shown in Fig 5.

In addition, farms in the municipality Someren had the highest in-degree and out-degree centrality. It is clear that the top three broiler nodes (municipalities) with the highest in-degree and out-degree centrality were the same with the municipalities with highest capacity (Someren, Peel en Maas and Hardenberg). Naturally, the farms with higher capacity usually have more consumers and suppliers and they are more likely to spread contaminations.

*Slaughterhouse.* The slaughterhouses had relatively high in-degree centrality; they receive products from broiler nodes from all directions. The maximum in-degree centrality was 51 (median was 11.5). On average, each slaughterhouse collected broilers from 14 broiler farms. In addition, 20 slaughterhouses provided products to 24 processors who do not have slaughtering business. On average each slaughterhouse sold slaughtered broilers to 16 processors without slaughtering business and max out-degree centrality was 23 (median was 19). The out-degree centralities of slaughterhouse were also relatively high. That is because the distribution of processors and slaughterhouses is such that most are concentrated in the middle parts of the Netherlands. Therefore, any pairs of nodes could have higher possibility to trade with each other.

*Processors.* Processors with slaughtering business had the highest out-degree and in-degree centrality. The maximum in-degree centrality was 146 (median was 16) and out-degree centrality was 377 (median was 53). Because of this, they are more prone to spread contaminations. In addition, these 41 processors sold broiler products to 380 municipalities. A processor on average sold processed meat to 114 municipalities. The processors who have slaughtering businesses had higher in-degree centrality than the processors without slaughtering business. That is because the processors with slaughter business received raw materials from 162 broiler farms nodes while the processor node without slaughter business bought slaughter meat from only 20 slaughterhouses. Each municipality on average purchased processed chicken meat from 18 of the 41 (44%) processors. The maximum in-degree centrality was 30 and the median was 18. Among 380 municipalities, Amsterdam, Rotterdam and the Hauge had the highest in-

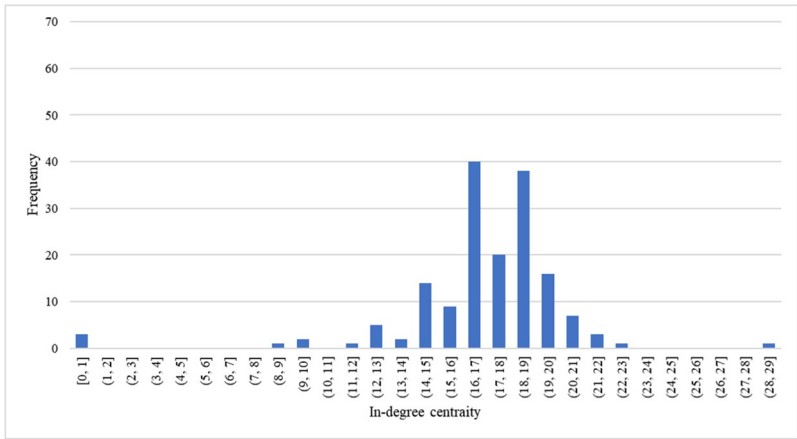

**Fig 6. Distribution of in-degree centrality of retailers (final consumers).**

degree centrality. Fig 6 shows that there is not much difference between in-degree centrality of final consumers, varying from 16 to 20. That is because almost 70% of suppliers (processors) were in the middle part of the Netherlands and having the similar capacity, so the processors may have the same possibility to deliver products to all municipalities.

*Importer and exporter.* 14 slaughterhouses (70%) exported slaughtered broiler meat to other counties and 6 of them (30%) imported raw materials from other counties. 14 processors (34%) exported products broiler meat to other countries and 4 of them (10%) had import ventures with other countries. The data shows that the actors with higher capacity have international trades with other countries.

**Betweenness centrality of involved actors.** Besides degree centrality, betweenness centrality of nodes is also relevant because a node with higher betweenness centrality would pass through more trade relations and would be at a higher risk to be exposed to contamination. Based on the distribution of 693 nodes in the network, 20 actors with highest betweenness centrality were found, including 18 processors and 2 broiler farms. The top 6 were processors with slaughtering business with a maximum betweenness centrality of 51,655. It also shows that 41% of all trade connections in Dutch poultry meat network passed through these six processing plants.

The actors with high degree centrality may not have high betweenness centrality. Betweenness centrality is based on the location of the actors. If the actors are in the middle stage of the supply chain, they tend to have higher degree centrality and betweenness centrality, such as processors. Because in general, the actors with higher degree centrality have more trade relations, there are more possibilities to pass through more potential paths between any two actors. However, if the actors are in the start or the end of the supply chain, they tend to have higher degree centrality but low betweenness centrality, such as final consumers. In addition, the trade distance was also analysed. The maximum distance between two linked nodes was 262km, the minimum distance was 0 km and the average distance was 80km. Almost 70% of the actors had a trade link with other nodes which were located within 100km. Compared to other subnetworks, the average distance from breeder farms to broiler farms was the lowest, namely 42km; from broiler farm to processor with slaughterhouse was the highest, namely 94km. The visualization of geographic distance between broiler breeder farms and broiler farms is shown in Fig 7 as an example.

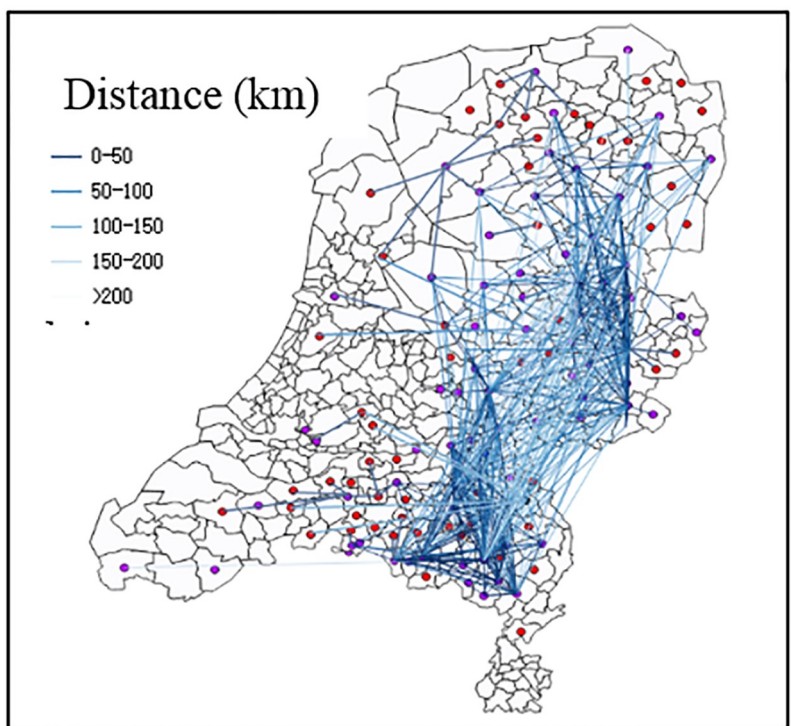

**Fig 7. An example of geographical distance representation from broiler breeder farms to broiler farms.**

**Trade density of involved actors.** Table 3 shows the trade density of each sub-network. Sub-network slaughterhouse—processors without a slaughterhouse had highest trade density. Among all pairs of nodes, 17% of them had trade links. That was also because the distribution of processors and slaughterhouses were concentrated in the middle so that they have higher possibility to trade with each other. The lowest density was in the subnetwork broiler farm-slaughterhouse, which was 0.9% of all directed edges. In the entire network, the trade relations represented 0.2% of total possible trade links.

To sum up, the broiler processors with slaughtering business had the highest centrality. They are prone to receiving contaminated broiler meat from broiler farms and spread

**Table 3. Trade density of sub-networks and entire network.**

|  | Broiler breeder farm-broiler farm | Broiler farm-slaughterhouse | Slaughterhouse-processors without a slaughterhouse | Broiler farm-processor with a slaughterhouse | Processors (all)-retailer/final consumers | Importer/exporter-slaughterhouse | Importer/exporter-processor (all) | Total net-work |
|---|---|---|---|---|---|---|---|---|
| **Nr. of nodes** | 252 | 182 | 44 | 179 | 421 | 22 | 43 | **693** |
| **Nr. of trade relations** | 1038 | 289 | 329 | 998 | 6592 | 20 | 19 | **9285** |
| **Trade density** | 1.6% | 0.8% | 17% | 3.1% | 3.7% | 4.3% | 1.0% | **1.9%** |

contaminated meat to consumers. The processor with slaughtering business is likely a crucial point in the broiler meat supply chain.

## Literature and expert validation

The results in this study showed that the processors with a slaughterhouse (or slaughterhouses with a processing facility) were more critical than farms in potential risk for contamination spread, which is consistent with the findings in literature. For example, Van Asselt and colleagues [30] analysed 20 thousand samples at various points of the poultry supply chain in the Netherlands from 2002 to 2005 and observed that, compared with broiler farms, slaughterhouses had more prevalence for *Campylobacter*. They found that a pre-slaughter *Campylobacter* prevalence of 20% to 31% and for breast skin samples a prevalence in the range of 10% to 40% over the years 2002 to 2005.

In a similar case, Jozwiak *et al*. [31] conducted a long study on the rate of infection with *Campylobacters* in a broiler meat supply chain in Hungary. Samples were collected from a broiler flock from the first day of life to the slaughter of the animals. In both summer and winter period, the days 0 to 12 chicks were not found to be contaminated. However, at day 42, they found *Campylobacter* spp. on every sampling point at the slaughterhouse. At the slaughter house, 93.3% of the live birds were infected with *Campylobacter* spp., and at the end of the processing line, the infection rate was 100%. This long-period study demonstrated that slaughterhouses were relatively critical.

Furthermore, a research conducted in Argentina [32] reported that the highest proportions (83.3%) of *Campylobacter* positive samples were observed in carcasses at retailers and 60% of *Campylobacter* cells were found in fecal samples from breeder farms. Only 3.3% of the samples in broiler farms were infected by *Campylobacter*. The proportions of *Campylobacter* positive carcasses at the slaughterhouses were 33.3%.

Overall, these cases support the view that the processors with a slaughterhouse constitute the most vulnerable step in the broiler meat supply chain. The results were checked with a domain expert of the Dutch poultry industry. The results were found to be consistent with the observations of the experts.

## Limitations and recommendations

In this study, much of the information about many of the actors, such as processers, was obtained from secondary sources, such as D&B Hoovers and Kompass websites. These data sources list all broiler slaughterhouses and processors in the Netherlands, but the capacity information of all actors was missing. Therefore, for some slaughterhouses an older capacity data (e.g. 2009) was extracted from the Poultry Site. Current capacities could thus be different from those in 2009. However, even if reliable and consistent trade data would be available, the estimate of the trade links depends on the quality of the estimate of the scaling factor $q_0$. To have a better estimate of the scaling factor, some of the actual trade links should be known. There were hundreds of actors in the network, but only few actors reveal their customers and suppliers. If more real trade links were known, a better estimate of $q_0$ can be made and a better trade network design, and thus a better traceability system, can be established. Furthermore, the gravity model considered only capacity and distance of actors. However, there were also other factors influencing possibility of trade. For instance, the existence of product hallmarks such as organic, affects the trade network.

The generalizability of these results is subject to certain limitations. First, the research only considered direct trade relations. In real trade network, there also exists indirect trades. For instance, when actors recall unqualified products from their consumers, they increase their in-

degree centrality. Also, the inter-trades between actors (horizontal trades between the same type of actors) were not considered. If these additional pathways were considered, the in-and-out degree centrality of some nodes and trade density of some sub-networks would increase.

Second, this research did not take trade frequency and temporal decisions into account. If a node delivers products or buys product frequently, then the in-and-out-degree centrality of this node would increase. In addition, temporal decisions are also major factors influencing number of trade relations. For example, if quality problems are found in some actors, then the contract would be quit temporally. Finally, in this research, the per capital consumption and the number of inhabitants was used as capacity of consumption. However, the consumption of broiler meat in each municipality cab be influenced by other factors such as age and religion. Furthermore, we should also consider some dynamic changes of the number and capacity of actors in the Dutch poultry industry. Pinior *et al*. [14] indicated that actors that are identified as food safety critical nodes today may not be that critical over a period of time, and vice versa. Therefore, the trade network should be updated regularly with new available data (e.g. capacity and location).

## Conclusion

In this paper, a Dutch broiler meat trade network was developed to identify critical points in the supply chain. The track-and-trace system was built using graph theory to identify the potential contamination sources and critical paths.

This study has found that generally the broiler processors with a slaughterhouse, with or without a processing facility, were the most critical points in the broiler meat supply chain. They are most prone to receive contaminated broiler meat from broiler farms and spread contaminated meat to consumers. The centrality of breeder farms and broiler farms was relatively low, and the most vulnerable farms in the network were from these municipalities: Someren, Peel en Maas and Hardenberg. These areas also had highest capacity in terms of the number of chicks. Among the 380 municipalities, the retailers from Amsterdam, Rotterdam and the Hague are more vulnerable to receiving contaminations.

## Supporting information

**S1 File.** [33–37].
(DOCX)

**S1 Data.**
(ZIP)

**S1 Fig.**
(TIF)

**S2 Fig.**
(TIF)

## Acknowledgments

We thank Mrs Marloes Heijne of Wageningen Bioveterinary Research for providing the source of data about the number of broiler farms and breeder farms per municipality and their capacities. We would like to thank Mr. Meng Lu, a PhD student at Wageningen University, for helping write the R script for calculating betweenness centrality. We would like to thank Mr. Stefan Maranus and Ms. Denise Kersjes, BSc and MSc student, respectively, at Wageningen University, for computing distances between municipalities. We would like to thank Mr. Maranus for

reviewing the article and editing references and cross-references, and would like to thank Ms. Kersjes for contacting many of the slaughterhouses and processors and requesting trade data; we would also like to thank the few companies that provided us the required data for our research.

## Author Contributions

**Conceptualization:** Hans Marvin.

**Data curation:** Ayalew Kassahun.

**Formal analysis:** Shuai Hao.

**Investigation:** Shuai Hao.

**Methodology:** Yamine Bouzembrak, Hans Marvin.

**Supervision:** Ayalew Kassahun.

**Writing – original draft:** Shuai Hao.

**Writing – review & editing:** Ayalew Kassahun, Yamine Bouzembrak, Hans Marvin.

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
