## [Decision Letter · Decision Letter 0]

14 Apr 2020

PONE-D-20-02018

Identification of potential vulnerable points and paths of contamination in the Dutch broiler meat trade network

PLOS ONE

Dear Dr. Kassahun,

Thank you for submitting your manuscript to PLOS ONE. After careful consideration, we feel that it has merit but does not fully meet PLOS ONE’s publication criteria as it currently stands. Therefore, we invite you to submit a revised version of the manuscript that addresses the points raised during the review process.

Manuscript lacks in the quality of preparation. I agree with reviewers, and authors should improve the manuscript. Please review the referee comments and make your peer revision.

We would appreciate receiving your revised manuscript by May 29 2020 11:59PM. To enhance the reproducibility of your results, we recommend that if applicable you deposit your laboratory protocols in protocols.io, where a protocol can be assigned its own identifier (DOI) such that it can be cited independently in the future. For instructions see: http://journals.plos.org/plosone/s/submission-guidelines#loc-laboratory-protocols

We look forward to receiving your revised manuscript.

Kind regards,

Arda Yildirim, Ph.D.

Academic Editor

PLOS ONE

Journal Requirements:

Additional Editor Comments:

This manuscript is well-designed work. It is necessary to improve the manuscript by examining the questions that need to be clarified in a way. For your guidance, you can check the reviewers' comments. Thank you for giving us the opportunity to consider your work.

Reviewers' comments:

Reviewer's Responses to Questions

**Comments to the Author**

1. Is the manuscript technically sound, and do the data support the conclusions?

Reviewer #1: Yes

Reviewer #2: Yes

2. Has the statistical analysis been performed appropriately and rigorously? 

Reviewer #1: Yes

Reviewer #2: Yes

3. Have the authors made all data underlying the findings in their manuscript fully available?

Reviewer #1: Yes

Reviewer #2: Yes

4. Is the manuscript presented in an intelligible fashion and written in standard English?

Reviewer #1: Yes

Reviewer #2: Yes

5. Review Comments to the Author

Reviewer #1: Authors described broiler poultry meat chain as a complex and therefore susceptible to many potential contaminations that may occur. To ensure a safe product for the consumer, an efficient traceability system is required that enables a quick and efficient identification of the potential sources of contamination and proper implementation of mitigation actions. In this study authors described and explored the use of graph theory to construct a food supply chain network for the broiler poultry meat supply chain in the Netherlands and tested it as a traceability system. The first steps which have been taken by the authors were building the graph, with the broiler breeder farms, broiler farms, slaughterhouses, processors, and retailers. Authors also described carefully the capacity data, production volume, trade volumes of each supply chain actor were gathered from various sources. The trade relationships between the supply chain actors were collected and the missing relationships were estimated using the gravity model. In addition, authors computed trade density to get insight into the complexity of sub networks. Author described the critical nodes at each stage of the Dutch broiler meat

supply chain and verified obtained results with a domain expert of the Dutch poultry industry

and literature. The results indicated that processors with own slaughtering facility were

the most critical points in the broiler meat supply chain. Modelling the broiler meat trade network have been presented by using QGIS software. The schematic representation of the analysed broiler meat supply chain and broiler meat trade network have been presented.

From my point of view the food and the presence of contaminants concerning work is an interesting compendium of making consumers aware of how easily it is possible to contaminate the Dutch broiler meat trade. Besides the cognitive value, the manuscript has meritum value. I am definitely in favor of publishing this manuscript in PLOS one journal.

Reviewer #2: The authors have built the track-and-trace system in Dutch broiler meat using graph theory to identify the potential contamination sources and critical paths. The manuscript is well structured and reports a very interesting study with valuable results, and some changes are suggested.

Minor comments:

Line 7: Number of affiliation is wrong

Line 20: modeled (UK past participle) should be changed to modeled (USA past participle)

Line 22: sub networks should be changed into sub-networks all through the manuscript.

Line 329: Pinior et al.: you miss the no. of reference

-The discussion needs to be streamlined and better ordered

6. PLOS authors have the option to publish the peer review history of their article (what does this mean?). If published, this will include your full peer review and any attached files.

Reviewer #1: Yes: Jowita Samanta Niczyporuk

Reviewer #2: Yes: Nagah Arafat

---

## [Author Response · Author response to Decision Letter 0]

28 Apr 2020

Response to reviewers

Below are extracted comments together with our answers. 

1. Is the manuscript technically sound, and do the data support the conclusions?

Reviewer #1: Yes

Reviewer #2: Yes

Response: Thank you. 

2. Has the statistical analysis been performed appropriately and rigorously?

Reviewer #1: Yes

Reviewer #2: Yes

Response: Thank you. 

3. Have the authors made all data underlying the findings in their manuscript fully available?

Reviewer #1: Yes

Reviewer #2: Yes

Response: Thank you. 

4. Is the manuscript presented in an intelligible fashion and written in standard English?

Reviewer #1: Yes

Reviewer #2: Yes

Response: Thank you. 

4. Is the manuscript presented in an intelligible fashion and written in standard English?

Reviewer #1: Yes

Reviewer #2: Yes

Response: Thank you. 

5. Review Comments to the Author

Reviewer 1: 

Authors described broiler poultry meat chain as a complex and therefore susceptible to many potential contaminations that may occur. To ensure a safe product for the consumer, an efficient traceability system is required that enables a quick and efficient identification of the potential sources of contamination and proper implementation of mitigation actions. In this study authors described and explored the use of graph theory to construct a food supply chain network for the broiler poultry meat supply chain in the Netherlands and tested it as a traceability system. The first steps which have been taken by the authors were building the graph, with the broiler breeder farms, broiler farms, slaughterhouses, processors, and retailers. Authors also described carefully the capacity data, production volume, trade volumes of each supply chain actor were gathered from various sources. The trade relationships between the supply chain actors were collected and the missing relationships were estimated using the gravity model. In addition, authors computed trade density to get insight into the complexity of sub networks. Author described the critical nodes at each stage of the Dutch broiler meat supply chain and verified obtained results with a domain expert of the Dutch poultry industry and literature. The results indicated that processors with own slaughtering facility were the most critical points in the broiler meat supply chain. Modelling the broiler meat trade network have been presented by using QGIS software. The schematic representation of the analysed broiler meat supply chain and broiler meat trade network have been presented.

From my point of view the food and the presence of contaminants concerning work is an interesting compendium of making consumers aware of how easily it is possible to contaminate the Dutch broiler meat trade. Besides the cognitive value, the manuscript has meritum value. I am definitely in favor of publishing this manuscript in PLOS one journal.

Response Reviewer 1: 

Thank you for the positive remark. 

Reviewer 2:

The authors have built the track-and-trace system in Dutch broiler meat using graph theory to identify the potential contamination sources and critical paths. The manuscript is well structured and reports a very interesting study with valuable results, and some changes are suggested.

Minor comments:

Line 7: Number of affiliation is wrong

Line 20: “modelled” (UK past participle) should be changed to “modeled” (USA past participle)

Line 22: “sub networks” should be changed into “sub-networks” all through the manuscript.

Line 329: Pinior et al.: you miss the no. of reference

-The discussion needs to be streamlined and better ordered

Response Reviewer 2: 

Thank you for the positive remark. With reference to minor comments, we acknowledge all 4 comments as valid comments and updated the manuscript accordingly. They can be seen in the revised manuscript with track changes. Please note that what used to be Line 329 in the original manuscript, is now Line 386 in the track-changes version of the revised manuscript, and Line 341 in the clean version of the revised manuscript.

In order to address the comment “The discussion needs to be streamlined and better ordered” we have made two set of changes in order to address the comment as described below:

- The “Results and Discussion” section of the original manuscript contained the following 6 sub sections; and they are now reduced to 3. The original sub sections were: 1) The structure of the Dutch broiler meat supply chain, 2) Broiler meat trade network, 3) Degree centrality, 4) Betweenness centrality of involved actors, 5) Trade density of involved actors, 6) Literature and expert validation. We have now introduced a new 3rd sub section called “Vulnerability of actors to spread contaminations” which now contains the previous 3 sections (“Degree centrality”, “Betweenness centrality of involved actors”, and “Trade density of involved actors”) as sub-sub sections. We have added text under the new section to explain how the 3 sub-sections (“Degree centrality”, etc) are related to “Vulnerability of actors to spread contaminations”. We believe that this will bring substantial improvement in streamlining the discussion.

- We have updated the “The structure of the Dutch broiler meat supply chain” sub-section of the “Results and Discussion” section to bring more clarity to results and subsequent discussion.

---

## [Editor Report · Decision Letter 1]

5 May 2020

Identification of potential vulnerable points and paths of contamination in the Dutch broiler meat trade network

PONE-D-20-02018R1

Dear Dr. Kassahun,

We are pleased to inform you that your manuscript has been judged scientifically suitable for publication and will be formally accepted for publication once it complies with all outstanding technical requirements.

With kind regards,

Arda Yildirim, Ph.D.

Academic Editor

PLOS ONE

https://www.researchgate.net/profile/Arda_Yildirim2

Additional Editor Comments (optional):

Thank you for responding to all comments and for revising the manuscript. Regards,
---

## [Editor Report · Acceptance letter]

6 May 2020

PONE-D-20-02018R1 

Identification of potential vulnerable points and paths of contamination in the Dutch broiler meat trade network 

Dear Dr. Kassahun:

I am pleased to inform you that your manuscript has been deemed suitable for publication in PLOS ONE. Congratulations! Your manuscript is now with our production department. 

With kind regards,

on behalf of

Dr. Arda Yildirim 

Academic Editor

PLOS ONE